# Bone mineral density and trabecular bone score in elderly type 2 diabetes Southeast Asian patients with severe osteoporotic hip fractures

**Linsey U. Gani**[1]*, **Kundan R. Saripalli**[1], **Karen Fernandes**[2], **Suet F. Leong**[2], **Koh T. Tsai**[2], **Pei T. Tan**[3], **Le R. Chong**[2], **Thomas F. J. King**[1]

**1** Department of Endocrinology, Changi General Hospital, Singapore, Singapore, **2** Department of Radiology, Changi General Hospital, Singapore, Singapore, **3** Clinical Trials and Research Unit, Changi General Hospital, Singapore, Singapore

* Linsey.u.gani@singhealth.com.sg

## Abstract

### Introduction

Studies show trabecular bone score (TBS) may provide information regarding bone quality independent of bone mineral density (BMD) in type 2 diabetes (DM2) patients. We analyzed our Southeast Asian severe osteoporotic hip fracture patients to study these differences.

### Methods

We conducted a retrospective cross-sectional analysis of subjects admitted to Changi General Hospital, Singapore with severe osteoporotic hip fractures from 2014–2017 who had BMD performed. Electronic records were reviewed and subjects were classified as having diabetes according to the WHO 2019 criteria. DM2 patients were classified according to their HbA1c into well controlled (HbA1c < 7%) and poorly controlled (HbA1c $\geq$ 7%) DM2.

### Results

Elderly patients with hip fractures present with average femur neck T scores at the osteoporotic range, however those with DM2 had higher BMD and TBS values compared to non DM2 patients. These differences were statistically significant in elderly women—poorly controlled elderly DM2 women with hip fracture had the highest total hip T-score (-2.57 ± 0.86) vs (-2.76 ± 0.96) in well controlled DM2 and (-3.09 ± 1.01) in non DM2 women with hip fracture, p < 0.001. In contrast, TBS scores were lower in poorly controlled DM2 women with hip fracture compared to well controlled DM2 women with hip fracture (1.22 ± 0.11) vs (1.24 ± 0.09), but these were still significantly higher compared to non DM2 women with hip fracture (1.19 ± 0.10), p < 0.001. In elderly men with hip fractures, univariate analysis showed no statistically significant differences in TBS or hip or LS BMD between those with poorly controlled DM2, well controlled DM2 and non DM2. The differences in TBS and BMD remained significant in all DM2 women with hip fractures even after adjustments for potential

**Data Availability Statement:** All relevant data are within the manuscript and its Supporting information files. Data cannot be shared publicly

because of potentially identifying and patient sensitive information. Restriction of data access to study team as approved by the SingHealth Institutional Review Board - IRB (CIRB Ref 2017/2563) Data are available from the Singhealth IRB (contact for request of access: irb@singhealth.com.sg).

**Funding:** The authors received no specific funding for this work.

**Competing interests:** The authors have declared that no competing interests exist.

confounders. Differences in TBS and BMD in poorly controlled DM2 men with hip fractures only became significant after accounting for potential confounders. However, upon inclusion of LS BMD into the multivariate model these differences were attenuated and remained significant only between elderly women with well controlled DM2 and non DM2 women with hip fractures.

## Conclusions

Elderly patients with DM2 and severe osteoporosis present with hip fractures at a higher BMD and TBS values compared to non DM2 patients. These differences were significant after adjustment for confounders in all DM2 women and poorly controlled DM2 men with hip fractures, TBS differences were attenuated with the inclusion LS BMD. Further studies are needed to ascertain differences in BMD and TBS in older Southeast Asian DM2 patients with variable glycemic control and severe osteoporosis.

## Introduction

Diabetes and osteoporosis are both major health challenges. The global prevalence of diabetes among adults over 18 years has risen from 4.7% in 1980 to 8.5% in 2014 [1]. Worldwide, 1 in 3 women as well as 1 in 5 men over the age of 50 years old will experience osteoporotic fractures [2]. Asians, especially South Asians are predisposed toward DM2 to a greater extent than Caucasians [3]. Singapore has a prevalence of DM2 at 10.5% which is higher than the world average of 8.8%, with estimates of prevalence rising to 15% in 2050 [4]. It is also projected that more than 50% of all osteoporotic fractures will occur in Asia by the year 2050 [5]. Studies have shown that patients with DM2 have a higher risk of fragility fracture, including a 40% to 70% increased hip fracture risk [6,7]. Taken together this implies a burgeoning epidemic of diabetes and fragility fractures, especially in Asia.

Fracture assessment in DM2 patients is complex. DM2 patients have a high fracture risk despite higher bone mineral density (BMD) results [8–11]. These have been attributed to wide ranging factors from types of medication use, presence of DM2 complications and disease duration [12,13]. Furthermore, studies have also shown that there are ethnic differences in the relationship between DM2 and fracture risk [14].

Trabecular bone score (TBS) is a grey–level textural metric that is obtained from lumbar spine dual energy X-ray absorptiometry (DXA) images. Decreased TBS has been found to be associated with an elevated risk for osteoporotic fractures independent of BMD in cohort studies. These results were confirmed by a recent meta-analysis of prospective cohort data [15] and adopted as evidence in position papers [16,17].

Various studies have looked into differences in TBS between DM2 and non-DM2 patients. The majority of studies in ethnic Caucasian populations have demonstrated the use of TBS independently of BMD in predicting lower bone quality in DM2 patients. However, studies in different ethnic groups have shown varying results particularly with respect to gender and age [18,19]. In particular, TBS differences were found in studies in those younger than 65 years old [20,21] and were not seen in the older population [22].

The variable performance of TBS in different population could be explained by the differential effects of clinical variables such as gender, age and osteoporotic hip T-score which has the greatest impact on TBS score in study subjects with DM2 [23]. Other studies have also noted

potential impact of body mass index (BMI) and body composition on TBS and lumbar spine (LS) BMD results in older men [24–27]. Furthermore, previous studies have also alluded to the impact of fasting plasma glucose, fasting insulin and HOMA-IR to TBS scores which points to the complexity of bone quality measurements in DM2 subjects [20,22].

Given the variable findings of the relationships of TBS and DM2 in different ethnicities and clinical groups, we sought to validate its performance in elderly Southeast Asian patients with severe osteoporosis. We hypothesized that severe osteoporotic elderly patients with DM2 will have lower bone trabecular scores despite having higher BMD values. In this study, we analyzed an older Southeast Asian cohort with severe osteoporosis presenting with fragility hip fractures in a regional hospital setting in Singapore to study the differences in BMD and TBS in patients with and without DM2.

## Subjects and methods

### Study population

We conducted a retrospective cross-sectional analysis of all patients admitted to Changi General Hospital with acute fragility hip fractures from 2014–2017. Clinical and demographic data were extracted from the electronic medical records. There were 1378 patients who were admitted for hip fractures during this period. We excluded 66 admissions for recurrent hip fractures and 174 patients who did not have any BMD performed. We also excluded patients with TBS or BMD reports with at least one lumbar level that were not included due to degeneration, instrumentation or previous fractures to minimize the impact of these conditions on LS BMD and TBS performance. Patients with previous exposure to bisphosphonates were also excluded from the analyses to reduce the impact of these drugs on BMD and TBS results. The final study population was 753 subjects. The cohort was divided into different gender and ethnicity according to the recorded ethnicity group and gender documented in the electronic medical records. The diagnosis of DM2 was established using the World Health Organization (WHO) 2019 criteria [28] on the basis of having an HbA1c of 6.5% or greater, or current treatment with oral antidiabetic drugs or insulin. We excluded patients with type 1 diabetes on the basis of the documented medical history of ketoacidosis and age of onset of diabetes before 25 years. DM2 patients were classified according to their HbA1c into well controlled (HbA1c < 7%) and poorly controlled (HbA1c $\geq$ 7%) DM2. The study was approved by the SingHealth Centralised Institutional Review Board for the waiver of informed consent in this retrospective data analysis.

### Clinical evaluation

Baseline clinical data was obtained on all patients presenting for fragility hip fracture. Information regarding medical history, including previous fragility fractures, parental history of fragility fractures, history of rheumatoid arthritis, steroid use of >5 mg/d for >3 months, presence of dementia, and previous amputation were collected from the electronic medical records. Medication data collected includes use of oral antidiabetic drugs, insulin doses (unit/kg), calcium and vitamin D supplementation, oral anti-resorptives and other anti-osteoporosis medications. We further recorded data on smoking status and alcohol intake (> 3 units/d). Anthropometric data was extracted from the electronic medical record and BMI was calculated as weight divided by height squared (kg/m$^2$).

### Biochemical evaluation

HbA1c level (%) was determined by immunoturbidimetric assay (Cobas 8000, Roche Diagnostics, Switzerland). HbA1C values within 6 months of admitted date were included in this

analysis, there were 9 DM2 women and 1 DM2 man without a recent HbA1C performed that was excluded from the DM2 subgroup comparison and regression analysis. Serum creatinine (umol/L) was measured using indirect ion-specific electrode (Roche Diagnostics, Switzerland) and estimated glomerular filtration rate (eGFR) was calculated using the Chronic Kidney Disease Epidemiology (CKD-EPI) equation. 25-hydroxyvitamin D was measured by radioimmunoassay (Roche Diagnostics, Switzerland), TSH and FT4 levels were also measured by immunoassay (Abbot affinity, Chicago, USA). Microalbuminuria was determined by an Albumin Creatinine Ratio (ACR) of 30–300 mg/g according to the American Diabetes Association guidelines. Patients with eGFR < 60 ml/min were classified to those with chronic kidney disease (CKD) for the purpose of this analysis.

## Bone mineral density and trabecular bone score measurements

All BMD scans were performed on a single densitometer (Hologic QDR Discovery Wi, USA). The region of interest (ROI) was set as the total hip (non-fractured hip site), femoral neck and first to fourth lumbar vertebrae. We excluded vertebrae with fractures or degeneration causing >1 standard deviation greater areal BMD from the immediately adjacent vertebrae in accordance with the International Society for Clinical Densitometry guidelines for individual vertebrae exclusion. The BMD precision error (percentage of coefficient variation) was 1% for the total hip with a least significant change of 0.034 g/cm$^2$, 2.3% for the femoral neck with a least significant change of 0.041 g/cm$^2$ and 1% for the lumbar spine with a least significant change of 0.022g/cm$^2$.

TBS was analysed with iNsight software (Version 3.0.2.0 Medimaps, France) for the same ROI used for BMD measurements by two of the authors (LRC and KF) who were blinded to the patient clinical data. TBS values were calibrated to standard values using the TBS calibration phantom (17 cm thickness and 25% fat mass equivalent), and were adjusted for BMI between 15–37 kg/m$^2$. No patients had a BMI above 35 in this study. The least significant change (LSC) for TBS was 4.24%. The short-term precision of TBS calculation was 1.53% (CV) from the same set of DXA scans used to evaluate the precision of the BMD measurements. TBS value of $\geq$ 1.35 is considered normal, between 1.20 to 1.35 intermediate, and $\leq$1.20 to be degraded.

## Statistical analysis

All analysis was performed using the Statistical Package for Social Science (SPSS version 21.0, Chicago IL, USA). Data were expressed as mean ± standard deviation (SD) for numerical data or frequency (percentage) for categorical data. For missing data, listwise deletion was applied. Sensitivity analysis was conducted to evaluate the robustness of the results in the presence of variables which had more than 20% of missing data. We further imputed data of patients with 3 available levels of LS spine for analysis to assess changes in our results and found consistency in the trend of the results. Interaction term analysis for gender and DM2 status were performed to ascertain the presence of gender differences in TBS and BMD differences. We included the interaction term analysis impact on TBS in S6 Table which found a significant interaction effect between gender and DM2 status.

In univariate analysis, numerical variables were compared with 2-sample T-test and categorical variables were examined with chi-square test or Fisher's exact test between DM2 and non-DM2 patients. We subdivided the DM2 groups into well controlled (HbA1C < 7%) and poorly controlled (HbA1C $\geq$ 7%) in comparing them to the non-DM2 patients. Multivariate analysis was used to assess the significant of differences in TBS and BMD measurements between well controlled and poorly controlled DM2 patients. In the first adjusted

model, we used analysis of covariance (ANCOVA) models adjusted for age and BMI (Model 1), further adjustments to this model was made to incorporate significant variables such as race, amputation, presence of CKD, 25(OH)D level (Model 2). A third model for analysis of differences in TBS was incorporated with addition of LS BMD to assess if TBS differences were independent of LS BMD value. A p-value of < 0.05 was considered statistically significant.

In a subgroup analysis of DM2 only patients, we analyzed the association between TBS and DXA measurements with treatment variables in well controlled and poorly controlled DM2 patients. The association of TBS and BMD were analyzed for associations with metformin, sulfonylurea, insulin use, presence of microvascular complication (defined by a history of microalbuminuria, renal impairment greater than CKD 3 and a previous history of amputation) and degree of DM2 control. Linear regression was performed to determine the relationship between diabetes status and TBS with other variables to be adjusted. A two-tailed, p-value of <0.05 was considered statistically significant.

## Results

### Cohort characteristics and demographics

There were 1138 patients admitted with fragility hip fractures from 2014 to 2017 who had BMD performed. After exclusion of patients with incomplete lumbar spine or TBS scores due to degeneration, unavailability of hip BMD due to previous instrumentation to contralateral hip and previous antiresorptive treatments, the final cohort analysed consisted of 753 patients (Fig 1). All patients had their BMD performed within 6 months of the fracture date with a median duration of 14–19 days from their hip fracture (S1 Table), there were no significant differences in duration between hip fracture date and BMD performed in DM2 and non DM2 patients. Of the hip fracture patients studied, 68.8% (n = 518) were women and 31.2% (n = 235) were men. 32.4% (n = 168) of the women had DM2 and 31.1% (n = 73) of the men had DM2. Median duration of diabetes was 8 years in DM2 women and 7 years in DM2 men. Average HbA1C was 7.10% in DM2 women and 7.40% in DM2 men. The majority of the patients in the cohort were of Chinese ethnicity, which is consistent with the general population demographic in Singapore.

Table 1 summarises the baseline characteristic differences of the patients with respect to gender and diabetes status. Women with DM2 who were admitted with fragility hip fractures were younger (76.9 years) compared to the non-DM2 (78.3 years) patients although this was not statistically significant. There was no statistically significant difference in age of men with hip fractures between DM2 and non-DM2 patients. Women with hip fractures who had DM2 had slightly higher BMI compared to non-DM2 patients (22.91 vs 21.75 kg/m$^2$). There was no significant difference in BMI in men between DM and non-DM patients. There were no significant differences in the rate of smoking, alcohol consumption, history of previous fractures, rheumatoid arthritis, steroid exposure, history of secondary osteoporosis and dementia between DM2 and non-DM2 patients. There was a higher rate of amputation in women and men with DM2. Average eGFR in DM2 patients were lower in both women and men, however the presence of CKD (as defined by eGFR <60) was statistically significantly higher only in DM2 women. 25-OH-D level was significantly lower in men with DM2 compared to non-DM2, there were no significant difference in the vitamin D status in women. There were higher rates of calcium and vitamin D supplementation use in DM2 women compared to non-DM2 women. There were no differences in length of stay and mortality between DM2 and non-DM2 women and men.

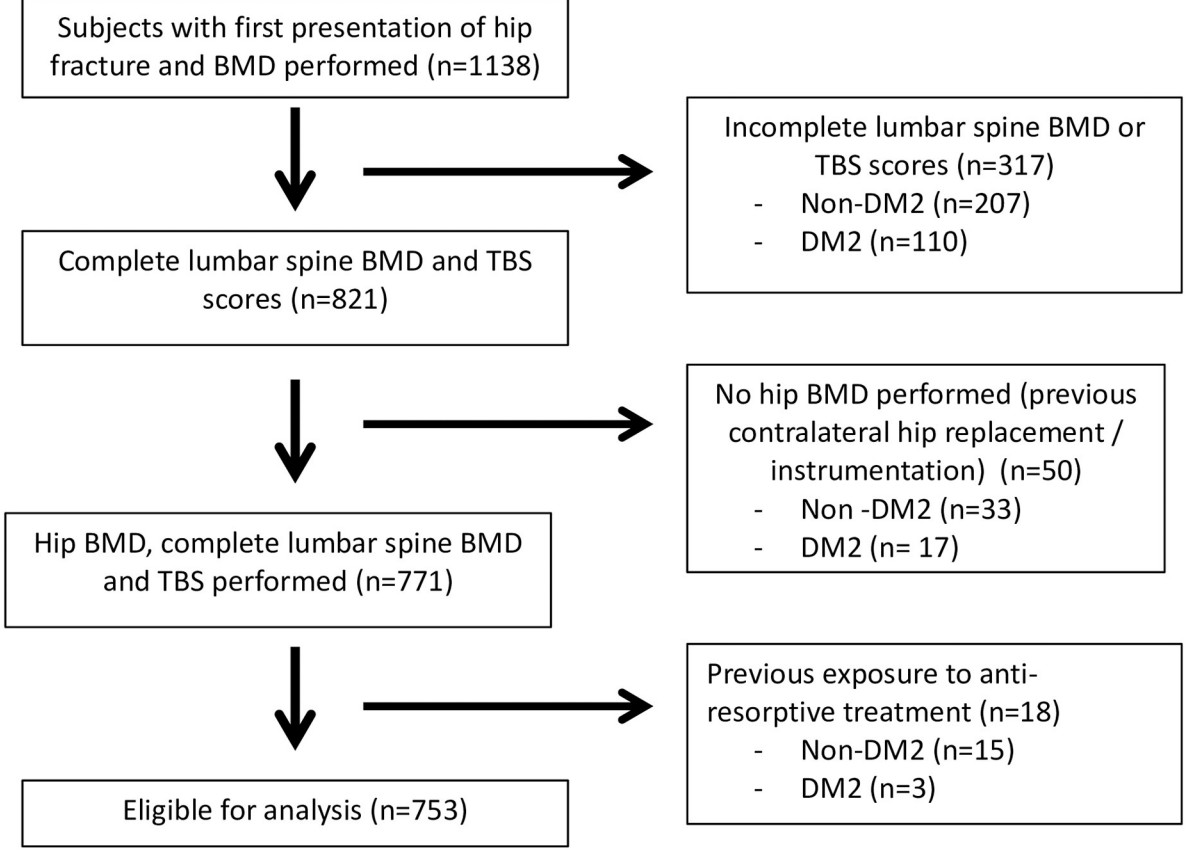

**Fig 1. Study subject inclusion and exclusion selection criteria.**

### TBS and BMD differences between women and men with well controlled (HbA1C < 7%) or poorly controlled DM2 (HbA1C ≥ 7%) compared to non DM2 patients with hip fractures

Table 2 summarizes TBS and BMD differences between women and men with well controlled or poorly controlled DM2 compared to non DM2 patients with hip fractures. Poorly controlled DM2 women present with hip fractures at younger age (73.7 ± 8.3 years) compared to those with well controlled DM2 (79.4 ± 7.8 years) and non DM2 women (78.3 ± 10.4 years) with hip fractures. This was similar in men with poorly controlled DM2 men presenting with hip fractures at younger age (71.4 ± 11.1 years) compared to well controlled DM2 (77.0 ± 9.0 years) and non DM2 men (74.1 ± 13.3 years) with hip fractures. Mean HbA1C in the well-controlled DM2 were 6.06 ± 0.65% and 6.18 ± 0.54% in elderly women and men respectively. In the poorly-controlled DM2 elderly women and men, average HbA1C were 8.29 ± 1.67% and 8.62 ± 1.74% respectively. Overall, there was a trend to higher BMD and T-score values in all sites as DM2 control worsened. The differences were significant in all sites in women, but was only significant in the total hip site in men.

TBS values were higher in well controlled DM2 women with hip fracture (1.22 ± 0.11) compared to poorly controlled DM2 women (1.24 ± 0.09), although both are in the category of intermediate degradation. This was also reflected in the lower LS BMD (0.82 ± 0.15 g/cm$^2$) vs (0.84 ± 0.15 g/cm$^2$) and T-score (-1.59 ± 1.35) vs (-1.36 ± 1.29) compared to those with well

**Table 1. Baseline demographic and clinical variables, trabecular bone score (TBS), bone mineral density (BMD) in patients who are non-diabetic (ND) and those with diabetes mellitus type 2 (DM2).** Patients with DM2 and non-DM2 (ND) are stratified by gender.

| Variable | Women (n = 518) | | | Men (n = 235) | | |
|---|---|---|---|---|---|---|
| Diabetes Status | ND (n = 350) | DM2 (n = 168) | p-value | ND (n = 162) | DM2 (n = 73) | p-value |
| Age (years) | 78.27 ± 10.39 | 76.90 ± 8.40 | 0.139 | 74.01 ± 13.27 | 74.16 ± 10.33 | 0.931 |
| Height (cm) | 150.26 ± 6.16 | 150.72 ± 5.95 | 0.422 | 160.37 ± 8.67 | 160.97 ± 7.98 | 0.613 |
| Weight (kg) | 49.16 ± 11.14 | 52.16 ± 10.36 | 0.004 | 57.02 ± 10.69 | 57.59 ± 11.25 | 0.710 |
| BMI (kg/m$^2$) | 21.75 ± 4.62 | 22.91 ± 4.09 | 0.006 | 22.24 ± 4.24 | 22.32 ± 4.61 | 0.894 |
| BMI group (Asian) | | | | | | |
| Underweight (< 18.5) | 91 (26.1%) | 19 (11.3%) | 0.002 | 29 (17.9%) | 14 (19.2%) | 0.994 |
| Normal (18.5–22.9) | 126 (36.1%) | 74 (44.0%) | | 67 (41.4%) | 30 (41.1%) | |
| Overweight (23–24.9) | 65 (18.6%) | 33 (19.6%) | | 33 (20.4%) | 14 (19.2%) | |
| Obese (≥ 25) | 67 (19.2%) | 42 (25.0%) | | 33 (20.4%) | 15 (20.5%) | |
| Race | | | | | | |
| Chinese | 278 (79.4%) | 110 (65.5%) | 0.002 | 131 (80.9%) | 47 (64.4%) | 0.028 |
| Malay | 50 (14.3%) | 34 (20.2%) | | 13 (8.0%) | 15 (20.5%) | |
| Indian | 9 (2.6%) | 14 (8.3%) | | 8 (4.9%) | 5 (6.8%) | |
| Others | 13 (3.7%) | 10 (6.0%) | | 10 (6.2%) | 6 (8.2%) | |
| Current smoker | 6 (1.7%) | 0 (0.0%) | 0.184 | 21 (13.0%) | 10 (13.7%) | 0.877 |
| Alcohol > 3 units per day | 3 (0.9%) | 0 (0.0%) | 0.544 | 9 (5.6) | 2 (2.7) | 0.510 |
| Previous fracture | 52 (14.9%) | 19 (11.3%) | 0.272 | 18 (11.1%) | 6 (8.2%) | 0.498 |
| Rheumatoid arthritis | 2 (0.6%) | 0 (0.0%) | 1.000 | 0 (0.0%) | 0 (0.0%) | NA |
| Steroids (> 3 months) | 3 (0.9%) | 1 (0.6%) | 1.000 | 2 (1.2%) | 0 (0.0%) | 1.000 |
| Dementia | 39 (11.1%) | 26 (15.5%) | 0.163 | 20 (12.3%) | 9 (12.3%) | 0.997 |
| Amputation | 1 (0.3%) | 8 (4.8%) | 0.001 | 0 (0.0%) | 2 (2.7%) | 0.096 |
| eGFR | 68.29 ± 22.05 | 59.61 ± 23.98 | <0.001 | 70.70 ± 23.52 | 61.77 ± 24.57 | 0.008 |
| CKD (eGFR < 60) | 131 (37.4%) | 83 (49.4%) | 0.010 | 48 (29.6%) | 29 (39.7%) | 0.127 |
| 25(OH)D (ug/L) | 21.97 ± 10.47 | 20.78 ± 10.97 | 0.241 | 25.15 ± 11.93 | 21.58 ± 9.67 | 0.026 |
| Calcium & Vitamin D supplementation | 96 (27.4%) | 64 (38.1%) | 0.014 | 21 (13.0%) | 16 (21.9%) | 0.081 |
| HbA1c (%) | 5.79 ± 0.87 | 7.10 ± 1.66 | <0.001 | 5.63 ± 0.77 | 7.40 ± 1.73 | <0.001 |
| Length of stay | 9 (6, 13) | 9 (7, 14) | 0.245 | 9 (7, 13) | 10 (7, 17) | 0.101 |
| Inpatient mortality | 0 (0.0%) | 2 (1.2%) | 0.105 | 3 (1.9%) | 0 (0.0%) | 0.554 |
| TBS | 1.19 ± 0.10 | 1.23 ± 0.10 | <0.001 | 1.31 ± 0.09 | 1.32 ± 0.10 | 0.229 |
| BMD Lumbar Spine(g/cm$^2$) | 0.72 ± 0.16 | 0.83 ± 0.15 | <0.001 | 0.87 ± 0.18 | 0.93 ± 0.18 | 0.126 |
| T score Lumbar Spine | -2.44 ± 1.33 | -1.47 ± 1.34 | <0.001 | -0.96 ± 1.55 | -0.57 ± 1.58 | 0.082 |
| BMD Total Hip(g/cm$^2$) | 0.56 ± 0.12 | 0.60 ± 0.11 | <0.001 | 0.69 ± 0.13 | 070 ± 0.13 | 0.402 |
| T score Total Hip | -3.09 ± 1.01 | -2.69 ± 0.92 | <0.001 | -2.43 ± 0.97 | -2.27 ± 1.02 | 0.251 |
| BMD Femur Neck(g/cm$^2$) | 0.47 ± 0.11 | 0.51 ± 0.10 | <0.001 | 0.57 ± 0.12 | 0.58 ± 0.13 | 0.812 |
| T score Femur Neck | -3.22 ± 0.89 | -2.83 ± 0.92 | <0.001 | -2.73 ± 0.90 | -2.69 ± 0.97 | 0.731 |

Numeric data was presented in mean ± SD, except LOS was presented in median (interquartile range).

controlled DM2, with all being in the same osteopenic category. Non DM2 women with hip fracture had the lowest TBS in the degraded category (1.19 ± 0.10) and LS BMD (0.72 ± 0.15 g/cm$^2$) with T-score at (-2.49 ± 1.35) in the osteoporotic category.

In men with hip fractures, those with poorly controlled DM2 had significantly higher total hip BMD and T-score (0.73 ± 0.12 g/cm$^2$, T-score -2.01 ± 0.88) compared with non DM2 men (0.69 ± 0.13 g/cm$^2$, T-score -2.43 ± 0.97).

**Table 2. TBS and BMD values in DM2 and non DM2 women and men stratified to well controlled (HbA1c < 7%) and poorly controlled (HbA1c ≥ 7%).**

| Variable | | Women (n = 509) | | | Men (n = 234) | | | |
|---|---|---|---|---|---|---|---|---|
| Diabetes Status | Non-DM (n = 350) | Hba1c < 7% (n = 85) | Hba1c ≥ 7% (n = 74) | p-value | Non-DM (n = 162) | HbA1c <7% (n = 36) | HbA1c ≥ 7% (n = 36) | p-value |
| Age | 78.27 ± 10.39 | 79.44 ± 7.77 | 73.64 ± 8.32 | <0.001 | 74.01 ± 13.27 | 76.94 ± 8.98 | 71.36 ± 11.08 | 0.163 |
| BMI | 21.75 ± 4.62 | 23.19 ± 4.00 | 22.70 ± 4.28 | 0.015 | 22.24 ± 4.24 | 22.10 ± 3.81 | 22.65 ± 5.34 | 0.844 |
| HbA1C (%) | | 6.06 ± 0.65 | 8.29 ± 1.67 | | | 6.18 ± 0.54 | 8.62 ± 1.74 | |
| TBS | 1.19 ± 0.10 | 1.24 ± 0.09 | 1.22 ± 0.11 | <0.001 | 1.31 ± 0.09 | 1.31 ± 0.10 | 1.34 ± 0.10 | 0.143 |
| BMD L-spine (g/cm$^2$) | 0.72 ± 0.16 | 0.84 ± 0.15 | 0.82 ± 0.15 | <0.001 | 0.89 ± 0.18 | 0.90 ± 0.16 | 0.95 ± 0.20 | 0.188 |
| BMD total Hip (g/cm$^2$) | 0.56 ± 0.12 | 0.59 ± 0.11 | 0.62 ± 0.10 | <0.001 | 0.69 ± 0.13 | 0.68 ± 0.14 | 0.73 ± 0.12 | 0.099 |
| BMD F-neck (g/cm$^2$) | 0.47 ± 0.11 | 0.50 ± 0.10 | 0.52 ± 0.10 | <0.001 | 0.57 ± 0.12 | 0.56 ± 0.13 | 0.60 ± 0.12 | 0.252 |
| T score Lumbar spine | -2.44 ± 1.33 | -1.36 ± 1.29 | -1.59 ± 1.35 | <0.001 | -0.96 ± 1.55 | -0.79 ± 1.41 | -0.33 ± 1.75 | 0.106 |
| T score Total Hip | -3.09 ± 1.01 | -2.76 ± 0.96 | -2.57 ± 0.86 | <0.001 | -2.43 ± 0.97 | -2.48 ± 1.08 | -2.01 ± 0.88 | 0.051 |
| T score F-neck | -3.22 ± 0.89 | -2.91 ± 0.92 | -2.68 ± 0.89 | <0.001 | -2.73 ± 0.90 | -2.89 ± 1.00 | -2.46 ± 0.90 | 0.129 |

We performed multivariate analysis to assess if differences in TBS and BMD in DM2 patients with well or poorly controlled DM2 versus non DM2 patients with hip fractures could be confounded by other potential variables. Covariates included in the analysis were variables that were found to be significantly different between DM2 and non DM2 patients which included age, BMI category, race, history of amputation, vitamin D status and presence of CKD. The results are shown in Table 3a for well controlled DM2 patients and Table 3b for poorly controlled DM2 patients.

Differences in TBS and BMD L spine remained statistically significant despite adjustments with covariates in well and poorly controlled DM2 women with hip fractures. TBS differences became attenuated upon inclusion of LS BMD into the model and remained only significantly different in well controlled DM2 women with hip fractures (Table 3a).

Differences in TBS and BMD at all sites were only significantly different in poorly controlled DM2 men with hip fractures after adjustments with covariates (Table 3b). However, these TBS differences became attenuated upon inclusion of LS BMD into the model and became non-significant in poorly controlled DM2 men with hip fractures.

## Association of TBS with LS BMD and BMI

S1 Table shows that there is a significant correlation between BMD and TBS values with the strongest correlation between BMD LS and TBS values. These were equally significant in both elderly men and women with or without DM2 who had hip fractures. These correlations remained significant with adjustments of age, BMI, vitamin D status and presence of CKD.

TBS results were also negatively correlated to BMI, although this did not reach statistical significance in both men and women. Conversely BMD was significantly positively correlated with BMI at all sites (p<0.05) for both men and women (S2 Table). Differences in TBS were significant in the BMI category of underweight (< 18.5 kg/m$^2$) for both men and women with DM2. In the normal BMI category (18.5–22.9 kg/m$^2$), only DM2 women showed significantly higher TBS values compared to non-DM2 women. These differences were not seen in the overweight and obese categories (S1 Fig).

**Table 3.** a. Mean Differences in Trabecular Bone Score (TBS) and Bone Mineral Density (BMD) at the Lumbar Spine and Hip between subjects with well-controlled DM2 (HbA1c <7.0%) and non-DM2. b. Mean Differences in Trabecular Bone Score (TBS) and Bone Mineral Density (BMD) at the Lumbar Spine and Hip between subjects with poorly controlled DM2 (HbA1c ≥ 7.0%) and non-DM2.

| | Women | | | Men | | |
|---|---|---|---|---|---|---|
| | *Mean Difference* | *95% CI* | *p-value* | *Mean Difference* | *95% CI* | *p-value* |
| **TBS** | | | | | | |
| Age and BMI category adjusted | 0.05 | 0.03, 0.07 | <0.001 | 0.002 | -0.03, 0.04 | 0.925 |
| Multivariate adjusted† | 0.05 | 0.03, 0.07 | <0.001 | -0.003 | -0.04, 0.03 | 0.857 |
| Multivariate adjusted^ | 0.02 | 0.001, 0.05 | 0.044 | -0.003 | -0.03, 0.03 | 0.834 |
| **BMD Lumbar Spine** | | | | | | |
| Age and BMI category adjusted | 0.10 | 0.07, 0.13 | 0.017 | 0.02 | -0.04, 0.08 | 0.543 |
| Multivariate adjusted† | 0.09 | 0.06, 0.13 | <0.001 | -0.0004 | -0.06, 0.06 | 0.990 |
| **BMD Total Hip** | | | | | | |
| Age and BMI category adjusted | 0.03 | 0.001, 0.05 | 0.042 | -0.003 | -0.05, 0.04 | 0.880 |
| Multivariate adjusted† | 0.03 | 0.002, 0.06 | 0.038 | -0.02 | -0.06, 0.03 | 0.514 |
| **BMD Femur Neck** | | | | | | |
| Age and BMI category adjusted | 0.02 | 0.001, 0.05 | 0.042 | -0.01 | -0.05, 0.03 | 0.680 |
| Multivariate adjusted† | 0.03 | 0.002, 0.05 | 0.033 | -0.02 | -0.06, 0.02 | 0.377 |
| **TBS** | | | | | | |
| Age and BMI category adjusted | 0.02 | -0.002, 0.05 | 0.065 | 0.03 | -0.002, 0.07 | 0.063 |
| Multivariate adjusted† | 0.03 | 0.0002, 0.05 | 0.048 | 0.04 | 0.002, 0.07 | 0.040 |
| Multivariate adjusted^ | 0.01 | -0.02, 0.03 | 0.678 | 0.01 | -0.02, 0.04 | 0.364 |
| **BMD Lumbar Spine** | | | | | | |
| Age and BMI category adjusted | 0.07 | 0.04, 0.11 | <0.001 | 0.06 | -0.001, 0.13 | 0.055 |
| Multivariate adjusted† | 0.07 | 0.03, 0.11 | <0.001 | 0.08 | 0.10, 0.15 | 0.028 |
| **BMD Total Hip** | | | | | | |
| Age and BMI category adjusted | 0.04 | 0.01, 0.06 | 0.011 | 0.04 | 0.0001, 0.09 | 0.049 |
| Multivariate adjusted† | 0.04 | 0.01, 0.06 | 0.014 | 0.05 | 0.004, 0.10 | 0.034 |
| **BMD Femur Neck** | | | | | | |
| Age and BMI category adjusted | 0.03 | 0.01, 0.05 | 0.012 | 0.02 | -0.02, 0.06 | 0.266 |
| Multivariate adjusted† | 0.03 | 0.004, 0.05 | 0.025 | 0.02 | -0.03, 0.06 | 0.481 |

†Model was adjusted for age, BMI category, race, amputation, eGFR < 60, 25(OH)D.

^ Model was adjusted for age, BMI category, race, amputation, eGFR < 60, 25(OH)D, BMD LS.

## Subgroup analysis in DM2 patients between well controlled vs poorly controlled DM2 patients with hip fractures

We analysed differences between well controlled and poorly controlled DM2 patients to elucidate potential contributors to differences in their BMD and TBS. There were significantly higher percentage of poorly controlled DM2 women on metformin (83.8%) vs those with well controlled DM2 (68.2%). There were also higher percentage of insulin use among poorly controlled DM2 women and men compared to those who are well controlled, however there were no significant difference in insulin unit per kg administered. We assessed for associations between the use of insulin, metformin or sulphonylurea, presence of microvascular complications, duration of DM2 and well controlled vs poorly controlled DM2 to TBS scores or BMD results in both women and men with DM2. Our analysis showed in DM2 women with hip fractures, poorer DM2 control was associated with lower TBS, whereas in men with hip fractures, poorer DM2 control was associated with higher TBS scores. We found that DM2 women with insulin use had significantly higher TBS score, lumbar spine and femoral neck

BMD after adjustment for confounding variables. Of note the total number of insulin users in our cohort were small at n = 29 (S5a Table). When we included LS BMD into analysis of TBS associations with DM2 variables, only LS BMD remained significantly associated with TBS implying that TBS associations with LS BMD was stronger than any DM2 variables (S5b Table). A separate analysis with HbA1c as a continuous variable did not show any significant association of HbA1C with TBS and BMD values. To reduce the probability of a skewed distribution of HbA1C affecting this result, we repeated the analysis with a log transformation of the HbA1c to reduce its variability, however the results were similar (S7 Table).

## Discussion

Our study adds to the growing body of literature on the characteristics of BMD and TBS in DM2 patients, especially in older patients with severe osteoporotic hip fractures of different ethnic backgrounds with well controlled or poorly controlled DM2. In this study, we looked at older Southeast Asian patients with severe osteoporosis who have sustained fragility hip fractures to assess differences in BMD and TBS in DM and non-DM2 patients. Elderly patients over 70 years of age are a major source of more than 70% of health care cost and contributes 45% to 60% of all major fractures [29]. In this regard our findings may be most applicable to the oldest and most severe of osteoporosis patients. The DM2 prevalence in this study of around 30% is consistent with our national data for an older population [30]. DM2 women on average had hip fractures at a younger age (76.9 years) compared to non-DM2 (78.3 years) despite higher BMD and TBS values. In our cohort, men present on average at a younger age (74.0 years) with hip fractures compared to women. Inpatient mortality and length of stay did not differ between DM2 and non-DM2 patients. Our results underscore current understanding in the limitation of BMD and TBS use in assessing bone quality in elderly patients with osteoporosis and DM2. Although all patients with hip fractures present with average femur neck (FN) T-scores at osteoporotic range, patients with DM2 and hip fractures had consistently higher BMD and TBS values compared to non DM2 patients with hip fractures. These differences were statistically significant in between non DM2 elderly women and well controlled or poorly controlled DM2 elderly women despite adjustment for potential significant confounders. In elderly men with hip fractures, these differences were only significant between non DM2 and poorly controlled DM2 after adjustments for microvascular complications and vitamin D status. Although other studies demonstrated that TBS could be a useful adjunct in assessing bone quality of DM2 patients, we found that in our cohort of older severe osteoporotic patients that differences in TBS values showed similar trend to BMD LS Spine values. Differences in TBS were not independent of LS BMD values in men and women regardless of the degree of DM2 control. Within DM2 patients, elderly women with poorly controlled DM2 had lower TBS values compared to those that were well controlled but this trend were reversed in elderly DM2 men. It is interesting to note that when we looked at HbA1C as a continuous variable, differences in TBS and BMD did not exhibit significant associations.

There may be a few explanations for the differences of the results observed in our study compared to previously reported studies in the literature. Ours is a much older population with severe osteoporosis with fragility hip fractures, significantly lower BMI and higher percentages of poorly controlled DM2 patients. The average age in the study population was 79 years for women and 74 years for men. Previous studies have shown an accelerated loss of BMD and TBS with age, with the rate of TBS decline after 65 years increasing by 50% [31]. As there are no current normative TBS data for healthy adults in Singapore, if we were to reference a geographically close South East Asian nation (Thailand), our TBS results are within the range of their normative data for population between 70–80 years old [25]. Normative TBS

values in the 70–80 y/o female were recorded at 1.218. The TBS value in this population of severe osteoporotic elderly female non DM2 patient was recorded at 1.19 and DM2 was 1.23. The same normative data in elderly (70–80 years) Thai male documented a TBS of 1.302, in our study of severe osteoporotic elderly males TBS were 1.31 in non DM2 and 1.32 DM2 men. Further larger and longitudinal studies should focus on impact of DM2 in this elderly population to understand these differences better.

In addition, in this study we found that HbA1C as a continuous variable did not exhibit any impact on TBS and BMD values. However, separation of patient analysis based on HbA1C of 7% into well controlled and poorly controlled DM2 found that TBS were significantly lower in women with poorly controlled DM2 but was persistently higher in men with poorly controlled DM2. Future studies could consider assessing different levels of glycemic control threshold and its impact on BMD and TBS measurements in women and men with DM2. It is also worth noting that differences between TBS, LS BMD and total hip BMD of poorly controlled DM2 men and non DM2 men became statistically significant only after accounting for these important clinical variables. These also further imply the need for future study of BMD and TBS differences in DM2 population to take into account the contribution of DM2 disease control and complications. These factors may also explain our differing results from a previous study in elderly Japanese men with average of 72.9 years which compared DM2 patient with average HbA1C of 6.5% to non DM2 patients and did not find any significant differences in TBS results [22]. This same study also further demonstrated the potential contributor of impaired glycemic control and insulin resistance Other potential explanation for the differences in the results of our study is the lower BMI of our elderly population (21.8 kg/m$^2$ in non-DM and 22.9 kg/m$^2$ in DM women, 22.2 kg/m$^2$ and 22.3 kg/m$^2$ in non-DM2 and DM2 men respectively). Other studies predominantly looked at relatively younger DM2 populations of average age around 65 years with higher BMI. The Manitoba [32] study cohort had BMI of DM2 and non-DM2 patients around 29.7 vs 26.7 kg/m$^2$, while another study from Korea [20] had average cohort ages between 62–66 years old with BMI of in women of 25.3 kg/m$^2$ (DM) vs 24.5 kg/m$^2$ (non-DM) and in men of 24.4 km/m$^2$ (DM) vs 23.5 kg/m$^2$ (non-DM). In the Vietnamese study the cohort average age was 60 years old in DM2 women with an average BMI of 25.0 kg/m$^2$ and 56 years old in DM2 men with an average BMI of 25.7 kg/m$^2$ [21]. Recent analyses have indicated that TBS is inversely related to BMI and abdominal fat, it may well be that our DM2 population of relatively lower BMI may not exhibit these effects in lowering TBS. This was also evident in our results that DM2 women and men with underweight BMI and DM2 women with normal BMI had significantly higher TBS scores than non-DM2, with no differences found in the overweight and obese categories.

Previous studies looking at the effects of insulin on BMD have shown differing results with some studies finding that exogenous insulin increases [33], reduces [34] or has a neutral effect on BMD [35,36] (S3 Table). Interestingly, although in our univariate analysis there were no demonstrable differences between TBS and BMD of DM2 women who are on insulin and non-insulin users, there appears to a statistically significant association of higher LS and femoral neck BMD after adjustments for age, BMI, disease duration, HbA1C, microvascular complications and other oral hypoglycaemia agents in the multivariate model. However, as our study is cross-sectional analysis with small number of insulin users, further observational follow-up study will be needed to understand the relationship between the use of insulin in DM2 and BMD in the elderly DM2 women.

Taken together, bone quality assessment in DM2 patients remains a complex issue. Differences in DM2 disease control and complication, together with age and BMI may contribute differently to the results of current bone quality measurement differences. These differences

were most significant in elderly DM2 hip fracture patients with poorly controlled DM2 and would be important for clinicians who are looking after elderly DM2 patients to be aware of.

Our study has a few strengths in particular our large number of diabetic patients and completeness of data with regards to DM2 duration, medications use and rates of documented microalbuminuria and history of amputation. Our study cohort also consisted of a uniquely older patients who presented with fragility fractures and in this regard our results would be applicable to patients at highest risk of fracture. We were also able to rule out other potential causes of secondary osteoporosis in our study and were also able to document 25(OH)D levels and data regarding previous treatment with anti-resorptives and supplementation with calcium and vitamin D. We excluded those with history of vertebral fractures, degeneration and surgical instrumentation from falsely elevating the BMD and TBS results. To further reduce the potential interference of previous anti-resorptive use in the BMD, we excluded these patients from our analysis as well.

The limitations of this study include the recruitment of only elderly hip fracture patients in our study. These patients have severe osteoporosis and as such our results may not be applicable to younger patients without severe osteoporosis. All BMD scans were performed on a single densitometer (Hologic QDR Discovery Wi, USA) and TBS was analysed with iNsight software (Version 3.0.2.0 Medimaps, France), previous studies have shown that this first version of TBS may be more significantly impacted by BMI compared to the newer version [37]. Hence whether our results may be applicable to the newer version of TBS is not known. There was also a lack of age matched control and local normative TBS data in the elderly population to allow further inference of the effect of DM2 and hip fracture on TBS and BMD results. We also had a lack of retinopathy data that would be important to be included for microvascular complications; however studies have shown that microalbuminuria is also a reliable indicator of diabetic retinopathy [38–40]. Our study also did not address the issue of falls and sarcopenia, both of which are important factors in the contribution of fracture risks in DM2 patients [41]. The cross-sectional nature of the study also cannot account for the potential dynamic changes in bone that occurs with progression of DM2. The maximum duration of DM2 in our patients was 9 years which did not allow us to study the effect of a longer duration of DM2 (>10 years) and its impact on TBS. We also did not have the glycemic status of the non-DM2 patients and would be unable to rule out patients with impaired fasting glucose or undiagnosed DM2 in the non-DM2 cohort.

In conclusion elderly patients with DM2 and severe osteoporosis present with hip fractures at a higher BMD and TBS values compared to non DM2 patients. These differences were significant after adjustment for confounders in all DM2 women and poorly controlled DM2 men with hip fractures, TBS differences were attenuated with the inclusion LS BMD. Further studies are needed to ascertain differences in BMD and TBS in older Southeast Asian DM2 patients with variable glycemic control and severe osteoporosis.

## Supporting information

**S1 Fig. Relationship of trabecular bone score (TBS) and body mass index (BMI) in women and men with and without DM2.**
(TIF)

**S1 Table. Duration (days) from date of admission to BMD analysis stratified by gender and DM2 status.**
(DOCX)

**S2 Table. Correlation between TBS and BMD in DM2 and non DM2 patients stratified by gender.**
(DOCX)

**S3 Table. Correlation between BMI with TBS and BMD in DM2 and nonDM2 patients stratified by gender.**
(DOCX)

**S4 Table. Demographic and clinical variables of DM2 patients subdivided into gender and DM2 control.**
(DOCX)

**S5 Table.** a: Relationship of trabecular bone score with diabetes mellitus (DM2) medications, complications and glycaemia control (β Coefficient). b: Relationship of Trabecular Bone Score adjusted for Lumbar Spine BMD with Diabetes Mellitus (DM2) Medication, Complication and Glycaemia Control (β Coefficient).
(DOCX)

**S6 Table. Interaction term analysis between TBS and DM2 status with gender.**
(DOCX)

**S7 Table. Relationship of trabecular bone score with diabetes mellitus (DM2) medications, complications and glycaemia control (β Coefficient).**
(DOCX)

## Acknowledgments

We thank the help of the Valued Care Hip Fracture Program in Changi General Hospital, Singapore for their help in providing us with patient data.

## Author Contributions

**Conceptualization:** Linsey U. Gani, Le R. Chong, Thomas F. J. King.

**Data curation:** Linsey U. Gani, Kundan R. Saripalli, Karen Fernandes, Suet F. Leong, Koh T. Tsai, Pei T. Tan, Le R. Chong.

**Formal analysis:** Linsey U. Gani, Pei T. Tan, Thomas F. J. King.

**Investigation:** Linsey U. Gani, Kundan R. Saripalli, Karen Fernandes, Suet F. Leong, Thomas F. J. King.

**Methodology:** Linsey U. Gani, Suet F. Leong, Pei T. Tan, Thomas F. J. King.

**Project administration:** Linsey U. Gani, Kundan R. Saripalli, Karen Fernandes, Koh T. Tsai, Thomas F. J. King.

**Software:** Karen Fernandes, Koh T. Tsai.

**Supervision:** Linsey U. Gani, Karen Fernandes, Suet F. Leong, Koh T. Tsai, Le R. Chong, Thomas F. J. King.

**Validation:** Kundan R. Saripalli, Karen Fernandes, Suet F. Leong, Koh T. Tsai, Pei T. Tan, Le R. Chong, Thomas F. J. King.

**Writing – original draft:** Linsey U. Gani, Thomas F. J. King.

**Writing – review & editing:** Linsey U. Gani, Kundan R. Saripalli, Karen Fernandes, Suet F. Leong, Koh T. Tsai, Pei T. Tan, Le R. Chong, Thomas F. J. King.

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
