## [Decision Letter · Decision Letter 0]

16 Jun 2020

PONE-D-20-14006

Bone Mineral Density and Trabecular Bone Score in Older Type 2 Diabetes Southeast Asian Patients with Osteoporotic Hip Fractures

PLOS ONE

Dear Dr. Gani,

Thank you for submitting your manuscript to PLOS ONE. After careful consideration, we feel that it has merit but does not fully meet PLOS ONE’s publication criteria as it currently stands. Therefore, we invite you to submit a revised version of the manuscript that addresses the points raised during the review process.

Reviewers are particularly concerned about notable features of your study population and the differences between them and other cohorts of patients with DM2 who have been studied regarding low trauma fracture.  In a revised MS, please be explicit about the unique features of your cohort and what conclusions, if any, might be applicable more generally.  How will reading this MS help either clinicians or investigators think about fracture risk in patients with DM?

We look forward to receiving your revised manuscript.

Kind regards,

Robert Daniel Blank, MD, PhD

Academic Editor

PLOS ONE

Journal Requirements:

2. In ethics statement in the manuscript and in the online submission form, please provide additional information about the patient records used in your retrospective study. Specifically, please ensure that you have discussed whether all data were fully anonymized before you accessed them and/or whether the IRB or ethics committee waived the requirement for informed consent. If patients provided informed written consent to have data from their medical records used in research, please include this information.

3. Please note that according to our submission guidelines (http://journals.plos.org/plosone/s/submission-guidelines), outmoded terms and potentially stigmatizing labels should be changed to more current, acceptable terminology. For example: “Caucasian” should be changed to “white” or “of [Western] European descent” (as appropriate).

4. Please include your tables as part of your main manuscript and remove the individual files. Please note that supplementary tables should remain as separate "supporting information" files.

5. Please include a caption for each figure.

Reviewers' comments:

Reviewer's Responses to Questions

**Comments to the Author**

1. Is the manuscript technically sound, and do the data support the conclusions?

Reviewer #1: Yes

Reviewer #2: Partly

Reviewer #3: Partly

2. Has the statistical analysis been performed appropriately and rigorously? 

Reviewer #1: Yes

Reviewer #2: Yes

Reviewer #3: Yes

3. Have the authors made all data underlying the findings in their manuscript fully available?

Reviewer #1: Yes

Reviewer #2: Yes

Reviewer #3: Yes

4. Is the manuscript presented in an intelligible fashion and written in standard English?

Reviewer #1: Yes

Reviewer #2: Yes

Reviewer #3: Yes

5. Review Comments to the Author

Reviewer #1: This study presents a detailed analysis of TBS, and lumbar spine, femoral neck BMD in a cohort of women and men of Asian origin with Diabetes compared to those without diabetes. The study is well conducted. However, the main limitation of the study is that the cohort analysed included only people with fracture. Thus, the contribution of TBS to fracture risk could not be assessed.

Comments:

Introduction is very long and has some very detailed descriptions for ethnic differences which are more suited for the Discussion section of the manuscript. I suggest reducing the Introduction and only presenting a succinct summary of the ethnic differences to build the hypothesis.

Methods: Some details related to study design are missing. For example, it is not clear whether hip fracture patients were invited for a clinical evaluation or their records were simply extracted from examination of medical records.

When was BMD measurement performed in relation to hip fracture (prior, after) and at what interval?

How was the duration of diabetes established?

Minor:

Figure 1 - should include the number of people with DM for all stages.

Reviewer #2: Bone mineral density (BMD) and trabecular bone score (TBS) assessed from DXA each provide independent information regarding skeletal status and fracture risk. Type 2 diabetes is increasingly recognized as a risk factor for fracture despite higher BMD, though the mechanism for this is complex. Reduced TBS in women with type 2 diabetes has been noted in many studies including an unreferenced meta-analysis (PMID 31214749) and may explain some of the excess fracture risk. The same meta-analysis did not detect altered TBS in men.

The current retrospective cross-sectional analysis was performed in women and men admitted with hip fracture who subsequently underwent DXA. Type 2 diabetes mellitus and covariates were identified from review of medical records. The authors found that TBS was higher (not lower) in DM2 women with hip fracture compared to those without DM2, and could not identify a TBS difference in men. BMD was increased at all skeletal sites in women with DM2 versus those without DM2; again no differences were observed in DM2 men.

General comments:

Overall, the report is well written and the analytical approach is appropriate. Increased BMD in DM2 is already widely known and this study contributes relatively little in that regard. The authors speculate on why their results differ from those reported elsewhere, and suggest this may relate to the “Asian diabetes phenotype” with a different pattern of low BMI and visceral adiposity, though this does not explain why their results differ from other Asian populations none of which showed higher TBS in DM2 women. Is it not possible that this reflects unique characteristics in the hip fracture population? The TBS implications of the study are therefore uncertain. In the absence of clinically relevant outcomes (fractures) the authors can do little more than conclude “Further studies are needed to ascertain the differences in BMD and TBS in Southeast Asian DM2 patients”.

Specific comments:

1. The number of individuals with hip fracture who did not undergo BMD testing is uncertain. This is important to report since if the included patients differ from those who were excluded this might bias results. Also the interval between hip fracture and DXA is not stated and needs to be clarified, since rapid BMD loss (especially from the hip) is known to occur following hip fracture. Finally, the reference data used for T-score reporting needs to be stated (including whether these are gender-neutral or gender-matched).

2. The study population excluded “patients with TBS or BMD reports with at least one lumbar level that were not included due to degeneration, instrumentation or previous fractures”. I am not sure if this is correctly worded or not, but exclusion of cases based upon a single vertebral level showing structural artifact would be very restrictive, especially in an elderly population with hip fractures where some degree of degenerative/structural change would be almost universal. Please clarify.

3. The authors state that they excluded individuals with previous bisphosphonate exposure. However bisphosphonates are widely used as treatment following hip fracture. This suggests either that treatment rates were extremely low (indeed only 18 individuals exclude due to antiresorptive treatment) or that the DXA testing was performed very shortly after hip fracture before treatment was initiated. Please clarify.

4. Approximately 1/3 of the hip fracture patients had DM2. Was this by design or does it reflect the true prevalence of diabetes in this population?

5. The authors do not adequately discuss the confounding effect that abdominal tissue thickness has on TBS measurements. “Raw” TBS decreases in relation to increasing abdominal tissue thickness and therefore the software algorithm includes a correction based upon BMI. This BMI adjustment may or may not be optimal for individuals with DM2, may not be applicable to populations that have a different pattern of abdominal adiposity, and differs for Hologic and GE DXA scanners (Hologic scanners are particularly sensitive to the effects of BMI.). Therefore, it is uncertain whether the reported TBS findings are a true reflection of skeletal properties or limitations in the TBS algorithm. A future version of the TBS algorithm that directly corrects for tissue thickness may be helpful, though this is not currently available for use.

6. The authors highlight differences between women and men in terms of how DM2 affects the BMD and TBS measurements. Significant differences were seen in women but not in men, this also reflects the relatively larger number of women versus men. Indeed, the 95% CIs for men in Table 2 (mean differences for TBS and BMD) appear to include the point estimates for women. It is therefore uncertain whether the gender difference is real or simply reduced power in men. A formal interaction analysis (gender x DM2) would clarify this.

7. Figure 2 is not very insightful and could be moved to the Supplementary section.

8. Please provide units for the coefficients reported in Supplementary Table 3.

9. Some references are repeated (#21 and #34, maybe others).

Reviewer #3: The manuscript is well written, topical and unique in the sense they have study severe OP patient (By hip fracture). It is however a pity that the study does not include non-T2DM-non-severe-OP matched for age controls. If you can add such controls, it will really add values to your study. If you cannot, then at least I would be comment on the value of TBS in severe OP with or without T2DM. Indeed, in both case, for example the Spine BMD corresponds to the osteopenic category while TBS corresponds to the degraded classification. Normal population at that age would have been in the partially degraded categories. Not giving such information is misleading as it give the wrong impression that TBS in such population (including T2DM) is normal while it is not. Along the same line, be careful in comparing your results with other studies as very few of them have Severe OP (by Hip fracture) and cannot be compared directly. Also to avoid any confusion you may add systematically “severe osteoporotic” e.g. non-DM2 vs DM2 severe osteoporotic patients.

Another very important point: While the outcomes contribute to a better understanding of the relationship between T2DM and TBS, the authors could go several steps further in the analysis to take into account current knowledge. Indeed, it has been repeatedly reported that TBS is lower in pre-T2DM patients or in uncontrolled T2DM patients as compared to controlled T2DM. Based on your supplementary table 2 you have a high number of patients with HbA1c patient above 7%. I would then repeat the analysis (and corresponding adjustment – tables 1&2) with three groups: non-DM2 vs DM2(HbA1c<7%) vs DM2(HbA1c>7%) severe osteoporotic patients.

Minor comments:

The mention of the study in HK population (ref 23) is coming unexpectedly as it is not on T2DM patient. What is the message you want to pass over here?

It might be worth mention in the introduction as stated above the impact of pre-T2DM and non-controlled T2DM on TBS. These results can better explained “variable performance of TBS in different population” than BMI as in most of the study BMI has been used as co-founding adjustment variable.

You are excluding patient with previous exposure to bisphosphonate. Is IT not the case for the other OP treatments? Does it mean that all the other patients with severe OP are not treated? Please explain.

Precision assessment for both BMD and TBS ae based on which population? Which age and number? If your CV for TBS is effectively 1% then your LSC should be 1.96 x root-mean-squared 2 x CV = 2.77% and not 4.24%. Can you explain the discrepancy?

The authors are performing multiple adjustment. Are you use that you are not over adjusting as many of the cofounding variables are not significantly different between groups …Wouldn’t you prefer to be more clinically strategic in the choice of the adjustment variable?

In table 2, for TBS it would also make an adjustment for Age, BMI and BMD lumbar spine, as we want to investigate the independence of TBS association between DM2 and non-DM2 from density.

Results / discussion:

See some of my major comments above.

I would still set the context where both non-DM2 and DM2 severe OP patients by hip fracture have degraded structure (TBS =< 1.23) while the spine BMD belongs to osteopenia category. Such results should be compared to expected BMD and TBS for that age (normal condition – reference curve).

You make a substantial paragraph on the impact of BMI on TBS while this one is barely 0.2% (while 17.6% for BMD). It seems to be that your results do not support the hypothesis that BMI would be a role here… You may reduce this one and focus on outcomes from new analysis to be performed (adjustment by spine BMD, < > HbA1c etc…)

Your report significant relationship between TBS and insulin for women in suppl. Table 3. But it is not mentioned at all in the text. On purpose?

In your limitations, I would clearly states that there is no age matched controls.

In conclusion, … I would add something along these lines: TBS and BMD in older DMs… are higher than non-DM2… in severe OP despite multiple adjustment. However overall values for hip BMD and spine TBS corresponds to osteoporosis and degraded structure respectively while spine BMD is osteopenic…

At the end, you can’t see that TBS is not useful in elderlies… you can only say that it may not be useful when you already have severe OP patients by Hip fracture…(although let’s see you results after adjustment for spine BMD and the category > HbA1c)

6. PLOS authors have the option to publish the peer review history of their article (what does this mean?). If published, this will include your full peer review and any attached files.

Reviewer #1: No

Reviewer #2: No

Reviewer #3: No

---

## [Author Response · Author response to Decision Letter 0]

24 Jul 2020

Dear PLOS ONE editor

We thank you for the opportunity to revise our manuscript and detail here our replies to the reviewers. We thank our reviewers for their time in reviewing this revision and hope this would be sufficient to answer their queries . 

Reviewer #1: This study presents a detailed analysis of TBS, and lumbar spine, femoral neck BMD in a cohort of women and men of Asian origin with Diabetes compared to those without diabetes. The study is well conducted. However, the main limitation of the study is that the cohort analysed included only people with fracture. Thus, the contribution of TBS to fracture risk could not be assessed.

Comments:

Introduction is very long and has some very detailed descriptions for ethnic differences which are more suited for the Discussion section of the manuscript. I suggest reducing the Introduction and only presenting a succinct summary of the ethnic differences to build the hypothesis.

We thank the reviewer for the comment and have amended our introduction and discussion sections accordingly. 

Methods: Some details related to study design are missing. For example, it is not clear whether hip fracture patients were invited for a clinical evaluation or their records were simply extracted from examination of medical records.

We thank the reviewer for the questions . This was a retrospective study conducted on hip fracture patients presenting to our institution between 2014-2017 , clinical and demographic details were extracted from their electronic medical records. We have clarified this in the descriptions of our methods

When was BMD measurement performed in relation to hip fracture (prior, after) and at what interval?

We thank the reviewer for this question. BMDs were performed during the same admission for hip fracture or within 6 months of the fracture , we have added this information into the manuscript as a supplementary table 1. The median is 14 to 19 days from date of admission from hip fracture to BMD analysis. 

How was the duration of diabetes established?

We thank the reviewer for the question. The diagnosis of DM2 was established using the World Health Organization (WHO) 2019 criteria (29) on the basis of having an HbA1c of 6.5% or greater, or current treatment with oral antidiabetic drugs or insulin. Duration of DM2 was established from the first documented diagnoses as established by the WHO definition to the date of hip fracture. These values were recorded according to the number of months, however for ease of reading these values were presented in terms of years in the analysis. We have added this information to further clarify this to the reader. 

Minor:

Figure 1 - should include the number of people with DM for all stages.

We thank the reviewer for the suggestion , this has been added to figure 1 .

Reviewer #2: Bone mineral density (BMD) and trabecular bone score (TBS) assessed from DXA each provide independent information regarding skeletal status and fracture risk. Type 2 diabetes is increasingly recognized as a risk factor for fracture despite higher BMD, though the mechanism for this is complex. Reduced TBS in women with type 2 diabetes has been noted in many studies including an unreferenced meta-analysis (PMID 31214749) and may explain some of the excess fracture risk. The same meta-analysis did not detect altered TBS in men.

The current retrospective cross-sectional analysis was performed in women and men admitted with hip fracture who subsequently underwent DXA. Type 2 diabetes mellitus and covariates were identified from review of medical records. The authors found that TBS was higher (not lower) in DM2 women with hip fracture compared to those without DM2, and could not identify a TBS difference in men. BMD was increased at all skeletal sites in women with DM2 versus those without DM2; again no differences were observed in DM2 men.

General comments:

Overall, the report is well written and the analytical approach is appropriate. Increased BMD in DM2 is already widely known and this study contributes relatively little in that regard. The authors speculate on why their results differ from those reported elsewhere, and suggest this may relate to the “Asian diabetes phenotype” with a different pattern of low BMI and visceral adiposity, though this does not explain why their results differ from other Asian populations none of which showed higher TBS in DM2 women. Is it not possible that this reflects unique characteristics in the hip fracture population? The TBS implications of the study are therefore uncertain. In the absence of clinically relevant outcomes (fractures) the authors can do little more than conclude “Further studies are needed to ascertain the differences in BMD and TBS in Southeast Asian DM2 patients”.

We thank the reviewer for his comments, we agree that the differences observed in our study is likely a reflection of the severe osteoporosis of this elderly hip fracture population. We further clarify in our discussion as suggested by reviewer #3 that although the TBS are higher in this study they still all belonged to the intermediate - degraded category which is consistent with LS BMD results obtained. We have also substantially added to the analysis in this revision by incorporating the differences in the well-controlled and poorly controlled DM2 patients as suggested by reviewer #3 and found that these differences were significant in all DM2 women and poorly controlled DM2 men. In particular the differences in poorly controlled DM2 men became significant after adjustments for microvascular complications and vitamin D status. We also further found that these differences were not independent of LS BMD results in this population. As such , we humbly believe that the finding of this study would further add to the current available literature and further inform clinicians in parts of the world looking after older DM2 patients with poorly controlled DM2 from a non-Caucasian ethnicity group on potential differences in TBS and BMD findings. We also suggest that future studies in DM2 patients should consider accounting for DM2 complications and vitamin D status 

Specific comments:

1. The number of individuals with hip fracture who did not undergo BMD testing is uncertain. This is important to report since if the included patients differ from those who were excluded this might bias results. Also the interval between hip fracture and DXA is not stated and needs to be clarified, since rapid BMD loss (especially from the hip) is known to occur following hip fracture. Finally, the reference data used for T-score reporting needs to be stated (including whether these are gender-neutral or gender-matched).

We thank the reviewer for this question , there were 174 patients who did not have BMD performed and were not included in this analysis. BMDs were performed during the admission or within 6 months of the hip fracture date. We included this information as supplementary table 1, median duration (days) from hip fracture date to BMD analysis was between 14-19 days and there were no significant differences between DM2 and non DM2 patients on the length of time from hip fracture and BMD analysis .

2. The study population excluded “patients with TBS or BMD reports with at least one lumbar level that were not included due to degeneration, instrumentation or previous fractures”. I am not sure if this is correctly worded or not, but exclusion of cases based upon a single vertebral level showing structural artifact would be very restrictive, especially in an elderly population with hip fractures where some degree of degenerative/structural change would be almost universal. Please clarify.

We thank the reviewer for the question , we excluded patients with incomplete BMD L spine and TBS scores to ensure that artefactual interferences from conditions such as osteoarthritis, severe scoliosis and degeneration would not interfere with the results of the analysis. We would like to assure the reviewer that when we performed the repeat analysis with inclusion of patients with at least 3 viable levels of LS Spine for BMD and TBS analysis , our results and analysis showed similar patterns to our current analysis. We have added this information into the statistical methods and sensitivity analysis to clarify this for our readers.

3. The authors state that they excluded individuals with previous bisphosphonate exposure. However bisphosphonates are widely used as treatment following hip fracture. This suggests either that treatment rates were extremely low (indeed only 18 individuals exclude due to antiresorptive treatment) or that the DXA testing was performed very shortly after hip fracture before treatment was initiated. Please clarify.

We thank the reviewer for the comment. Indeed osteoporosis treatment rate in the elderly in our community is very low , we have previously published osteoporosis treatment rates even after a hip fracture at 1 year was between 10-31% . In this study population about 10% of the patients have had a previous history of fracture. We hope to be able to improve these rates of treatment with the establishment of our osteoporosis liaison service and further implement primary fracture prevention treatments prior to the occurrence of a hip fracture. 

4. Approximately 1/3 of the hip fracture patients had DM2. Was this by design or does it reflect the true prevalence of diabetes in this population?

Singapore has one of the highest incidence of DM2 in the world and the DM2 prevalence in the elderly has been documented in the national statistics to be around 30% which is in keeping with this study population ( National Health Survey 1998, 2004, 2010, Ministry of Health, Singapore ), we have included this reference to clarify this point (Phan TP, Alkema L, Tai ES, Tan KHX, Yang Q, Lim W-Y, et al. Forecasting the burden of type 2 diabetes in Singapore using a demographic epidemiological model of Singapore. BMJ Open Diabetes Research & Care. 2014;2 )

5. The authors do not adequately discuss the confounding effect that abdominal tissue thickness has on TBS measurements. “Raw” TBS decreases in relation to increasing abdominal tissue thickness and therefore the software algorithm includes a correction based upon BMI. This BMI adjustment may or may not be optimal for individuals with DM2, may not be applicable to populations that have a different pattern of abdominal adiposity, and differs for Hologic and GE DXA scanners (Hologic scanners are particularly sensitive to the effects of BMI.). Therefore, it is uncertain whether the reported TBS findings are a true reflection of skeletal properties or limitations in the TBS algorithm. A future version of the TBS algorithm that directly corrects for tissue thickness may be helpful, though this is not currently available for use.

We thank the reviewer for the comments, we agree that further study on the impact of body composition especially abdominal tissue thickness on TBS is important . And we agree that in keeping with previous published studies the current Hologic TBS version are particularly sensitive for BMI. We have put this as a limitation of the study in the discussion section to clarify this point 

6. The authors highlight differences between women and men in terms of how DM2 affects the BMD and TBS measurements. Significant differences were seen in women but not in men, this also reflects the relatively larger number of women versus men. Indeed, the 95% CIs for men in Table 2 (mean differences for TBS and BMD) appear to include the point estimates for women. It is therefore uncertain whether the gender difference is real or simply reduced power in men. A formal interaction analysis (gender x DM2) would clarify this.

We thank the reviewers for the comment .Gender is known to have an effect on BMD and TBS as demonstrated by previous studies and hence we have separated the analysis of this study into different gender. We have added the information on gender and DM2 analysis into the supplementary section ( Supplementary table 6 ) of the manuscript. Our analysis have found that there is an interaction effect between gender and DM2 . With using Male with DM2 as reference group , female without DM2 were found to have significantly lower TBS . Female with DM2 also had statistically significant lower TBS compared to male with DM2 and there is no significant difference between male without DM2 and male with DM2. This is consistent with the results of the current analysis and multivariate model. Differences were only found in elderly males with poorly controlled DM2 and after adjustments with potentially confounding variables as related in the manuscript. 

7. Figure 2 is not very insightful and could be moved to the Supplementary section.

We thank the reviewer for the comment and have moved this figure into the supplementary section as supplementary figure 1

8. Please provide units for the coefficients reported in Supplementary Table 3.

We thank the reviewer for the comment . Supplementary table 3 is now supplementary table 5 and units for the covariate insulin (unit/kg ) , DM2 duration ( years ) and HbA1C ( % ) has been provided. 

9. Some references are repeated (#21 and #34, maybe others).

We thank the reviewer for the comment and have update the references appropriately to remove any duplicates 

Reviewer #3: The manuscript is well written, topical and unique in the sense they have study severe OP patient (By hip fracture). It is however a pity that the study does not include non-T2DM-non-severe-OP matched for age controls. If you can add such controls, it will really add values to your study. If you cannot, then at least I would be comment on the value of TBS in severe OP with or without T2DM. Indeed, in both case, for example the Spine BMD corresponds to the osteopenic category while TBS corresponds to the degraded classification. Normal population at that age would have been in the partially degraded categories. Not giving such information is misleading as it give the wrong impression that TBS in such population (including T2DM) is normal while it is not. Along the same line, be careful in comparing your results with other studies as very few of them have Severe OP (by Hip fracture) and cannot be compared directly. Also to avoid any confusion you may add systematically “severe osteoporotic” e.g. non-DM2 vs DM2 severe osteoporotic patients.

We thank the reviewer for this very insightful comment. We agree that our population studied belong to the “severe osteoporotic” group and we have clarified this further in our manuscript to avoid misleading the readers that the TBS values are normal. We have further inserted information regarding TBS degradation severity to clarify this point within the manuscript. We followed the software generated TBS degradation score and classification which was TBS value of � 1.35 is considered normal, 1.20 to 1.35 is considered to intermediate and � 1.20 to be degraded.

Another very important point: While the outcomes contribute to a better understanding of the relationship between T2DM and TBS, the authors could go several steps further in the analysis to take into account current knowledge. Indeed, it has been repeatedly reported that TBS is lower in pre-T2DM patients or in uncontrolled T2DM patients as compared to controlled T2DM. Based on your supplementary table 2 you have a high number of patients with HbA1c patient above 7%. I would then repeat the analysis (and corresponding adjustment – tables 1&2) with three groups: non-DM2 vs DM2(HbA1c<7%) vs DM2(HbA1c>7%) severe osteoporotic patients.

We thank the reviewer for the insightful comments and suggestions , we have repeated the analysis with the well- controlled and poorly controlled DM2 patients and comparing them to the non DM2 patients. We found that differences in TBS and BMD were significant in all DM2 women and persisted after adjustment for potential significant confounders. TBS and BMD differences became significant in elderly poorly controlled DM2 men after adjusting for microvascular complications and vitamin D status. Differences in TBS became attenuated after inclusion of BMD LS but remained significant only in elderly well controlled DM2 women. We hope that these results will help better inform future clinicians looking after elderly DM2 patients in interpreting TBS and BMD findings. We further suggest the importance of including details on DM2 microvascular complications and vitamin D status for future analysis as this may confound BMD and TBS results as shown in our elderly men with poorer DM2 control. 

Minor comments:

The mention of the study in HK population (ref 23) is coming unexpectedly as it is not on T2DM patient. What is the message you want to pass over here?

We thank the reviewer for this comment and agree that this reference may not provide additional information for the purpose of this study and have removed this paragraph and reference from the manuscript 

It might be worth mention in the introduction as stated above the impact of pre-T2DM and non-controlled T2DM on TBS. These results can better explained “variable performance of TBS in different population” than BMI as in most of the study BMI has been used as co-founding adjustment variable.

We thank the reviewer for this comment , we have added the references to studies which alluded to the role of impaired fasting glucose , insulin resistance and HOMA IR to TBS in addition to BMI in the introduction section. 

You are excluding patient with previous exposure to bisphosphonate. Is IT not the case for the other OP treatments? Does it mean that all the other patients with severe OP are not treated? Please explain.

We thank the reviewer for this comment. Indeed osteoporosis treatment rate in the elderly in our community is very low , we have previously published osteoporosis treatment rates even after a hip fracture at 1 year was between 10-31% . In this study population about 10% of the patients have had a previous history of fracture. We hope to be able to improve these rates of treatment with the establishment of our osteoporosis liaison service and better implement primary fracture prevention treatments prior to the occurrence of a hip fracture. 

Precision assessment for both BMD and TBS ae based on which population? Which age and number? If your CV for TBS is effectively 1% then your LSC should be 1.96 x root-mean-squared 2 x CV = 2.77% and not 4.24%. Can you explain the discrepancy?

We thank the reviewer for the question. We apologise for the error Our TBS CV are 1.53% with an LSC of 4.24% for this Hologic devices . This was a value calculated together with the assistance of our vendor. This has been amended appropriately in the manuscript. The BMD T score is based on our normative local population . TBS scores are not graded with a T score as we don’t currently have a local normative data to compare this to. We hope that future studies in our local population will be able to provide further information on the normative TBS data. 

The authors are performing multiple adjustment. Are you use that you are not over adjusting as many of the cofounding variables are not significantly different between groups …Wouldn’t you prefer to be more clinically strategic in the choice of the adjustment variable?

- Shall we adjust with only significant variables on top of age , BMI, CKD status ( yes or no ) , vitamin D , DM2 status ( Can I check how many non DM2 patients had documented HbA1C levels ? If this number is high , we could use HbA1C instead of DM2 status for adjustment ) , and BMD LSpine ( only for differences in TBS DM2 patients ) 

We thank the reviewer for this comment , we reduced the number of covariates in our multivariable analysis to the suggested covariates that are found to be significant in our univariate analysis. We have limited our variables analysed in the model to be age, BMI , race, presence of CKD , amputation and vitamin D status. 50% of our non-DM2 patients have HbA1C documented during the time of admission . As not all our non-DM2 patients had HbA1C recorded, we have opted to use DM2 status in our adjustments. 

In table 2, for TBS it would also make an adjustment for Age, BMI and BMD lumbar spine, as we want to investigate the independence of TBS association between DM2 and non-DM2 from density.

- could we also do this for the supplementary 3 table for TBS fully adjusted multivariate model ( ie including the BMD L Spine in the model )

We thank the reviewer for the insightful comment and have added this into a third model in our adjustments to assess the independence of TBS association between DM2 and non DM2 patients. We found that TBS difference were not independent of LS BMD differences in both well controlled and poorly controlled DM2 elderly patients. We have included this findings into our results and discussions. 

Results / discussion:

See some of my major comments above.

I would still set the context where both non-DM2 and DM2 severe OP patients by hip fracture have degraded structure (TBS =< 1.23) while the spine BMD belongs to osteopenia category. Such results should be compared to expected BMD and TBS for that age (normal condition – reference curve).

We thank the reviewer for this comment , we have inserted the TBS category for degradation - TBS value of � 1.35 is considered normal , � 1.20 to � 1.35 is considered to intermediate and �1.20 to be degraded. As we do not have a current normative data for our population, setting this in the context of a recent normative TBS and BMD study from a South East Asian population in Thailand ( Sritara et al Age-Adjusted Dual X-ray Absorptiometry-Derived Trabecular Bone Score Curve for the Lumbar Spine in Thai Females and Males JCD 2016 ; 19) , the value of TBS in this population is compared to the recorded value in the 70-80 y/o female TBS at 1.218. The TBS value in this population of severe osteoporotic elderly female non DM2 patient was recorded at 1.19 and elderly DM2 was 1.23. The same normative data in elderly (70-80 y/o) Thai male was documented in the study at TBS of 1.302, our study demonstrated TBS value of 1.31 in elderly non DM2 male and 1.32 in elderly DM2 male. We have added that future studies to better understand the role of TBS and BMD in our elderly population and normative data would be important to inform future studies in our population. 

You make a substantial paragraph on the impact of BMI on TBS while this one is barely 0.2% (while 17.6% for BMD). It seems to be that your results do not support the hypothesis that BMI would be a role here… You may reduce this one and focus on outcomes from new analysis to be performed (adjustment by spine BMD, < > HbA1c etc…)

We thank the reviewer for the comment and have shortened this discussion and moved the figure to the supplementary section of the manuscript. 

Your report significant relationship between TBS and insulin for women in suppl. Table 3. But it is not mentioned at all in the text. On purpose?

We thank the reviewer for the comment , as this was a cross sectional study we were not able to infer causality and simply stated the association as demonstrated in the multivariate analysis. Furthermore , we acknowledged the limitation of the small number of DM2 patients on insulin within this study ( n=29 ) as a limitation of the study. We cited previous studies on insulin and its effect in BMD .Exogenous insulin was found in various studies to increase, reduces or having a neutral effect on BMD. We added that it would be important for future longitudinal and larger study to better understand the relationship of exogenous insulin on BMD or TBS. 

In your limitations, I would clearly states that there is no age matched controls.

We thank the reviewer for the comment , we have added this important limitation point into our discussion. 

In conclusion, … I would add something along these lines: TBS and BMD in older DMs… are higher than non-DM2… in severe OP despite multiple adjustment. However overall values for hip BMD and spine TBS corresponds to osteoporosis and degraded structure respectively while spine BMD is osteopenic…

We thank the reviewer for the comment. We have amended our conclusion to state the pertinent findings in our re-analysis. We concluded that older DM2 patients presents with hip fractures at a higher TBS and BMD value compared to the non-DM2 patients. And these differences persist despite adjustment for covariates in DM2 women and poorly controlled DM2 men. We have further expounded on the differences in LS BMD, T score and Total hip T score in elderly men and women with DM2 and non DM2 in our discussions to clarify the corresponding values to the osteopenic or osteoporotic ranges . 

At the end, you can’t see that TBS is not useful in elderlies… you can only say that it may not be useful when you already have severe OP patients by Hip fracture…(although let’s see you results after adjustment for spine BMD and the category > HbA1c)

We thank the reviewer for the comment. We agree that the results of this study is not applicable to all elderly patients and hence have limited our report to the findings of this study - which are the significantly higher TBS and BMD values in elderly patients with DM2 who present with hip fractures. However, women with poorly controlled DM2 had lower TBS values compared to those that were well controlled but this trend were reversed in DM2 men. It is interesting to note that when we looked at HbA1C as a continuous variable, differences in TBS and BMD did not exhibit significant associations. These may imply the need for future study of BMD and TBS differences in DM2 population to take into account the contribution of DM2 disease control and complications

We mentioned in our discussion caution to our readers to limit this finding to the elderly hip fracture population and suggested that further studies would be important in understanding these differences better in this population. 

Corresponding Author

Linsey Gani

---

## [Decision Letter · Decision Letter 1]

14 Aug 2020

PONE-D-20-14006R1

Bone Mineral Density and Trabecular Bone Score in Elderly Type 2 Diabetes Southeast Asian Patients with Severe Osteoporotic Hip Fractures

PLOS ONE

Dear Dr. Gani,

Thank you for submitting your manuscript to PLOS ONE. After careful consideration, we feel that it has merit but does not fully meet PLOS ONE’s publication criteria as it currently stands. Therefore, we invite you to submit a revised version of the manuscript that addresses the points raised during the review process.

Please address the suggestions made by reviewer 3.  Thank you for the substantial changes you have made to address the initial critique of your MS.

We look forward to receiving your revised manuscript.

Kind regards,

Robert Daniel Blank, MD, PhD

Academic Editor

PLOS ONE

Reviewers' comments:

Reviewer's Responses to Questions

**Comments to the Author**

1. If the authors have adequately addressed your comments raised in a previous round of review and you feel that this manuscript is now acceptable for publication, you may indicate that here to bypass the “Comments to the Author” section, enter your conflict of interest statement in the “Confidential to Editor” section, and submit your "Accept" recommendation.

Reviewer #1: All comments have been addressed

Reviewer #3: All comments have been addressed

2. Is the manuscript technically sound, and do the data support the conclusions?

Reviewer #1: Yes

Reviewer #3: Yes

3. Has the statistical analysis been performed appropriately and rigorously? 

Reviewer #1: Yes

Reviewer #3: Yes

4. Have the authors made all data underlying the findings in their manuscript fully available?

Reviewer #1: Yes

Reviewer #3: Yes

5. Is the manuscript presented in an intelligible fashion and written in standard English?

Reviewer #1: Yes

Reviewer #3: Yes

6. Review Comments to the Author

Reviewer #1: Thank you for addressing my comments. I think the manuscript has improved. I have no further comments.

Reviewer #3: The paper has been significantly improved.

Please add on table 2 the age and BMI for each categories.

It can be misleading sometime in the text… when you are saying “elderly patients with DM2 and severe OP present with Hip fracture…. Compared to non DM2 patients”. It could almost applied that the non DM2 patients are not OP with hip fracture. So please throughout the manuscript, abstract and table make it clear….In patients with severe osteoporosis present with Hip fracture: DM2 vs non DM2

The fact that in poorly controlled women the TBS is lower than well controlled is now consistent with literature…. and clearly have clinical implication. This findings is already important and could be also highlighted in the abstract. Maybe by linking that with the results in men dilute the message. You may want to separate the sentence in two.

The fact that HbA1C as continuous variable does not work as good as category could be related to the non-gaussian distribution of the parameter (skewed)…. The log should then be used… (possible explanation….)

7. PLOS authors have the option to publish the peer review history of their article (what does this mean?). If published, this will include your full peer review and any attached files.

Reviewer #1: No

Reviewer #3: No

---

## [Author Response · Author response to Decision Letter 1]

9 Sep 2020

Dear Plos One Editor

We thank you for the opportunity to revise our manuscript and detail here our replies to the reviewers. We thank our reviewers for their time in reviewing this revision and hope this would be sufficient to answer their queries . 

Reviewer #3: The paper has been significantly improved.

Please add on table 2 the age and BMI for each categories.

Thank you for the comment , we have added age and BMI into table 2

It can be misleading sometime in the text… when you are saying “elderly patients with DM2 and severe OP present with Hip fracture…. Compared to non DM2 patients”. It could almost applied that the non DM2 patients are not OP with hip fracture. So please throughout the manuscript, abstract and table make it clear….In patients with severe osteoporosis present with Hip fracture: DM2 vs non DM2

Thank you for the comment , we have amended these accordingly to make it clear that these are DM2 and non DM2 patients with hip fracture . 

The fact that in poorly controlled women the TBS is lower than well controlled is now consistent with literature…. and clearly have clinical implication. This findings is already important and could be also highlighted in the abstract. Maybe by linking that with the results in men dilute the message. You may want to separate the sentence in two.

Thank you for the comment, we have amended this in the abstract and hope that this clarifies the point better 

The fact that HbA1C as continuous variable does not work as good as category could be related to the non-gaussian distribution of the parameter (skewed)…. The log should then be used… (possible explanation….)

Thank you for the comment and suggestion, we re-analyzed the multivariable regression using a log-transformation of the HbA1C and found no association of Log HbA1C with TBS. This is included in supplementary table 7. We have included this explanation in the body of the text to clarify this point.

---

## [Decision Letter · Decision Letter 2]

19 Oct 2020

Bone Mineral Density and Trabecular Bone Score in Elderly Type 2 Diabetes Southeast Asian Patients with Severe Osteoporotic Hip Fractures

PONE-D-20-14006R2

Dear Dr. Gani,

We’re pleased to inform you that your manuscript has been judged scientifically suitable for publication and will be formally accepted for publication once it meets all outstanding technical requirements.

Kind regards,

Robert Daniel Blank, MD, PhD

Academic Editor

PLOS ONE

Additional Editor Comments (optional):

Reviewers' comments:

Reviewer's Responses to Questions

**Comments to the Author**

1. If the authors have adequately addressed your comments raised in a previous round of review and you feel that this manuscript is now acceptable for publication, you may indicate that here to bypass the “Comments to the Author” section, enter your conflict of interest statement in the “Confidential to Editor” section, and submit your "Accept" recommendation.

Reviewer #1: All comments have been addressed

Reviewer #3: All comments have been addressed

2. Is the manuscript technically sound, and do the data support the conclusions?

Reviewer #1: Yes

Reviewer #3: (No Response)

3. Has the statistical analysis been performed appropriately and rigorously? 

Reviewer #1: Yes

Reviewer #3: (No Response)

4. Have the authors made all data underlying the findings in their manuscript fully available?

Reviewer #1: Yes

Reviewer #3: (No Response)

5. Is the manuscript presented in an intelligible fashion and written in standard English?

Reviewer #1: Yes

Reviewer #3: (No Response)

6. Review Comments to the Author

Reviewer #1: Thank you for addressing reviewers' comments. The manuscript benefited from the revision. I have no further comments.

Reviewer #3: (No Response)

7. PLOS authors have the option to publish the peer review history of their article (what does this mean?). If published, this will include your full peer review and any attached files.

Reviewer #1: No

Reviewer #3: No

---

## [Editor Report · Acceptance letter]

22 Oct 2020

PONE-D-20-14006R2 

Bone Mineral Density and Trabecular Bone Score in Elderly Type 2 Diabetes Southeast Asian Patients with Severe Osteoporotic Hip Fractures 

Dear Dr. Gani:

I'm pleased to inform you that your manuscript has been deemed suitable for publication in PLOS ONE. Congratulations! Your manuscript is now with our production department. 

Kind regards, 

on behalf of

Dr Robert Daniel Blank 

Academic Editor

PLOS ONE